# Floquet group theory and its application to selection rules in harmonic generation

Ofer Neufeld[1,2], Daniel Podolsky[2] & Oren Cohen[1,2]

Symmetry is one of the most generic and useful concepts in science, often leading to conservation laws and selection rules. Here we formulate a general group theory for dynamical symmetries (DSs) in time-periodic Floquet systems, and derive their correspondence to observable selection rules. We apply the theory to harmonic generation, deriving closed-form tables linking DSs of the driving laser and medium (gas, liquid, or solid) in (2+1)D and (3+1)D geometries to the allowed and forbidden harmonic orders and their polarizations. We identify symmetries, including time-reversal-based, reflection-based, and elliptical-based DSs, which lead to selection rules that are not explained by currently known conservation laws. We expect the theory to be useful for ultrafast high harmonic symmetry-breaking spectroscopy, as well as in various other systems such as Floquet topological insulators.

[1] Solid State Institute, Technion—Israel Institute of Technology, 32000 Haifa, Israel. [2] Physics Department, Technion—Israel Institute of Technology, 32000 Haifa, Israel. Correspondence and requests for materials should be addressed to O.N. (email: ofern@tx.technion.ac.il) or to O.C. (email: oren@technion.ac.il)

Symmetry has been used as a principle concept throughout science[1]. It simplifies problems and hints to initial ansatz solutions. For example, symmetries give rise to conserved quantities, and to selection rules for electronic transitions in molecules and solids[2]. A unique class of symmetries are spatio-temporal symmetries (denoted dynamical symmetries (DSs)), which are exhibited by time-periodic Floquet systems. Floquet systems are very widespread, including Floquet topological insulators[3–6], Floquet–Weyl semimetals[7,8], modulated photonic and solid state lattices[3,9–14], Bose–Einstein condensates[15–17], and more. Still, the manifestation of DSs and their observed selection rules in Floquet systems has not yet been formulated by a general approach, but rather has been limited to several ad-hoc cases[18–20].

A particular Floquet process of interest is harmonic generation (HG) in both the perturbative[21–23], and non-perturbative high harmonic generation (HHG) regime[24,25]. HHG provides a unique table-top source of coherent radiation in extreme UV and X-ray spectral regions, and serves for producing attosecond pulses for ultrafast spectroscopy[25]. The HG process is greatly affected by DSs in the driving laser and material target that dictate the allowed emission. For instance, HG by a half-wave symmetric driving laser that interacts with isotropic media results in odd-only harmonics[18]. Additionally, discrete rotational DSs have been employed for generating circularly polarized high harmonics using bi-circular two-color laser fields[26–34], as well as other combinations of drivers and molecular targets[19,20,35–38]. Notably, to date, all known selection rules in HG can be explained and derived from DSs, or equivalently by conservation laws.

Mathematically, symmetries are best described by group theory, which provides a rigorous platform for their analysis. Space–time groups that describe DSs (excluding time-reversal symmetries) were presented in the 1960s[39], yet they were not utilized for exploring Floquet systems. A closely related theory was developed in the 1970s describing so-called line-groups[40,41], which characterize the spatial symmetries of polymers. The selection rules in polymers are determined by analyzing the system's eigenstates and transition probabilities under finite perturbations. This approach however, is inappropriate for Floquet systems because they are continuously excited by a strong time-dependent perturbation that requires a dynamical theory. Selection rules in Floquet systems thus arise from the actual time-dependent dynamics, which is generally unknown and may be extremely complex (as in many-body systems). A unified analytical approach for describing DSs and their resulting selection rules in Floquet systems will therefore be useful in many areas of physics and chemistry.

Here, we explore the symmetries in Floquet systems using group theory. We systematically derive DSs as generalized products of spatial and temporal transformations in both $(2 + 1)$D and $(3 + 1)$D, yielding closed-form dynamical groups that describe the symmetries of Floquet systems. We prove that if a time-dependent Hamiltonian commutes with a dynamical group, then the group's generating operators dictate the evolution of all the physical observables in the system, and we derive the resulting restrictions for all DSs. We apply the theory to HG from gaseous, liquid, or solid media, in either collinear or non-collinear geometries (in both perturbative and non-perturbative regimes), and derive tables linking the harmonic emission selection rules to the symmetry of the laser–matter system. We introduce several DSs that lead to selection rules, including time-reversal-based DSs, reflection and inversion-based DSs, and an elliptical DS that can be used to control the polarization properties of high harmonics. Remarkably, some of the selection rules are not explained by currently known conservation laws.

## Results

**Symmetry elements in Floquet systems**. The symmetries of Floquet systems can be described by adjoining point group-like dimensions (molecular-like) that describe spatial symmetries, and an infinite and periodic dimension which is space group-like (lattice-like) that describes the $T$-periodic time axis and temporal symmetries. For simplicity, we exclude spatial transla-tional symmetries (extension to time-crystals is possible), and only deal with vectorial time-dependent functions (though the method also applies to scalar functions). Considering a general time-dependent vector field $\mathbf{E}(t)$, a symmetry of the field is an operation that leaves it invariant. Thus, $\hat{X}$ is a symmetry of $\mathbf{E}(t)$ if $\hat{X} \cdot \mathbf{E}(t) = \mathbf{E}(t)$. $\hat{X}$ can be a purely spatial operation, a purely temporal operation, or a product of temporal and spatial operations. In the $(2 + 1)$D case, there are only two relevant types of spatial symmetry elements: rotations, denoted by the operator $\hat{R}_n$ ($\hat{R}_n$ stands for rotation by an angle $2\pi/n$), and reflections, denoted by the operator $\hat{\sigma}$. In $(3+1)$D we also consider spatial inversion $\hat{i}$, and improper rotations $\hat{s}_n = \hat{\sigma}_h \cdot \hat{R}_n$ ($\hat{\sigma}_h$ stands for reflection in the plane perpendicular to the adjoined rotation axis). Temporal symmetry elements include time-reversal, denoted by $\hat{T}$ (which is order 2), and time-trans-lations, where translations by time $T/n$ are denoted by $\hat{\tau}_n$ (which is order $n$).

We derive here a general theory for Floquet systems by considering all products of spatial and temporal operations that close under group multiplication. We exclude purely spatial transformations[2] since they arise only in trivial cases (such as reduced dimensionality), where the symmetry group is a direct product of the spatial and dynamical groups. This exclusion greatly reduces the amount of DSs that permit closure. For instance, consider a general DS with temporal and spatial parts, $\hat{X}_t \cdot \hat{X}_s$. If this DS is raised to the order of $\hat{X}_t$, then $\hat{X}_s$ raised to the order of $\hat{X}_t$ must be the spatial identity (otherwise, we get a purely spatial transformation). This, combined with the fact that $T$ is the basic period of $\mathbf{E}(t)$, implies that either $\hat{X}_t$ and $\hat{X}_s$ have the same order, or $\hat{X}_s$ is the spatial identity, permitting purely temporal DSs. In what follows, this approach is used to systematically derive all DSs in Floquet groups according to their operation order.

**DSs and groups in $(2+1)$D**. We start out by considering the $(2 + 1)$D case, and derive DSs that are products of 2D spatial and temporal operations. Adjoining order-2 temporal operators with spatial reflection yields the following DSs (Fig. 1):

$$\hat{D} = \hat{T} \cdot \hat{\sigma} \tag{1}$$

$$\hat{Z} = \hat{\tau}_2 \cdot \hat{\sigma} \tag{2}$$

$$\hat{H} = \hat{T} \cdot \hat{\tau}_2 \cdot \hat{\sigma} \tag{3}$$

The same temporal operations can also be adjoined to rotations by 180° (Fig. 2):

$$\hat{C}_2 = \hat{\tau}_2 \cdot \hat{R}_2 \tag{4}$$

$$\hat{Q} = \hat{T} \cdot \hat{R}_2 \tag{5}$$

$$\hat{G} = \hat{T} \cdot \hat{\tau}_2 \cdot \hat{R}_2 \tag{6}$$

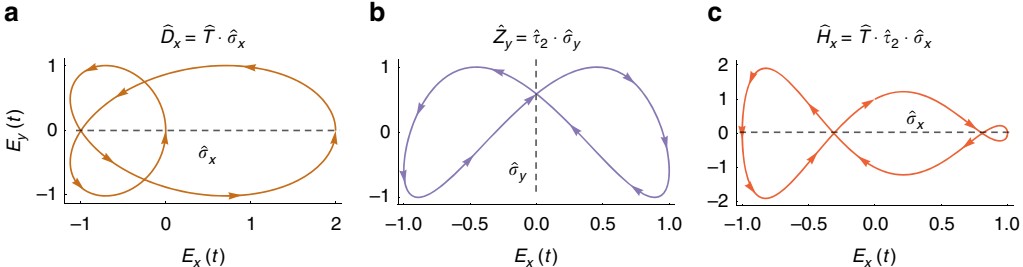

**Fig. 1** Order-2 spatiotemporal DSs in (2+1)D involving spatial reflection with examples for each symmetry. (**a**) $\hat{D}_x$ symmetry for the example field $\mathbf{E}(t)=$ $(\cos(\omega t)+\cos(2\omega t))\hat{x}+\sin(2\omega t)\hat{y}$, (**b**) $\hat{Z}_y$ symmetry for the example field $\mathbf{E}(t)=\sin(\omega t)\hat{x}+\sin(2\omega t+\pi/5)\hat{y}$, and (**c**) $\hat{H}_x$ symmetry for the example field $\mathbf{E}(t)$ $=\sin(\omega t)\hat{x}+\cos(3\omega t)\hat{y}$. The fields are represented on Lissajou plots. The spatial parts of the operators is indicated by dashed lines, colored arrows in the plots indicate the direction of time

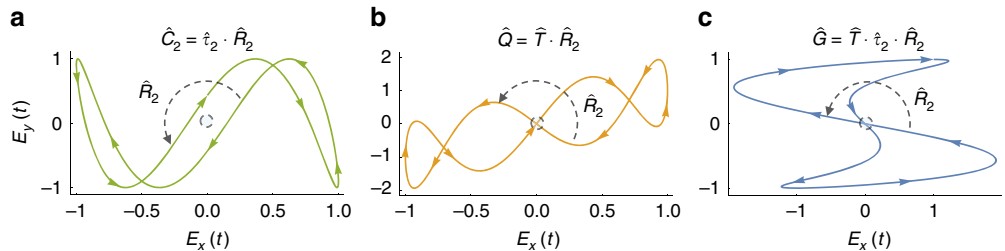

**Fig. 2** Order-2 spatiotemporal DSs in (2 + 1)D involving spatial rotations by 180° with examples for each symmetry. (**a**) $\hat{C}_2$ symmetry for the example field $\mathbf{E}(t)=\sin(\omega t)\hat{x}+\sin(3\omega t+\pi/7)\hat{y}$, (**b**) $\hat{Q}$ symmetry for the example field $\mathbf{E}(t)=\sin(\omega t)\hat{x}+(\sin(\omega t)+\sin(4\omega t))\hat{y}$, and (**c**) $\hat{G}$ symmetry for the example field $\mathbf{E}(t)$ $=(\sin(2\omega t)+\cos(3\omega t))\hat{x}+\cos(\omega t)\hat{y}$. The fields are represented on Lissajou plots. The spatial parts of the operators is indicated by dashed arrows, colored arrows in the plots indicate the direction of time

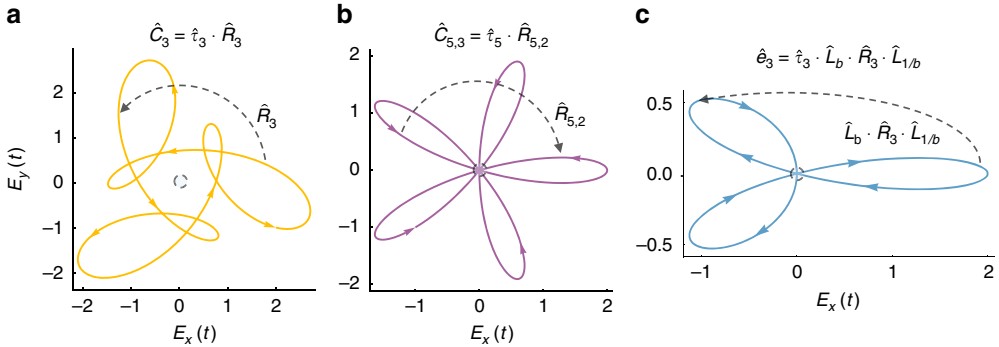

**Fig. 3** High-order spatiotemporal DSs in (2+1)D with examples of each symmetry. (**a**) $\hat{C}_3$ symmetry for the example field $\mathbf{E}(t)=(\cos(\omega t)+\cos(2\omega t)+\sin(4\omega t))\hat{x}+(\sin(\omega t)-\sin(2\omega t)-\cos(4\omega t))\hat{y}$, (**b**) $\hat{C}_{5,3}$ symmetry for the example field $\mathbf{E}(t)=(\cos(3\omega t)+\cos(2\omega t))\hat{x}+(\sin(3\omega t)-\sin(2\omega t))\hat{y}$, and (**c**) $\hat{e}_3$ symmetry for the example field $\mathbf{E}(t)=(\cos(\omega t)+\cos(2\omega t))\hat{x}+b(\sin(\omega t)-\sin(2\omega t))\hat{y}$. The fields are represented on Lissajou plots. The spatial parts of the operators is indicated by dashed arrows, colored arrows in the plots indicate the direction of time

Next, we map out higher order operators that involve rotations (temporal screw-axes):

$$\hat{C}_n = \hat{\tau}_n \cdot \hat{R}_n \qquad (7)$$

The operator $\hat{C}_n$ is a DS of bi-circular EM fields, and of the Hamiltonians of circularly polarized electric fields interacting with rotationally invariant molecules[19,29,30,32,33,38] (Fig. 3a). The $\hat{C}_n$ operator is generalized to cases where the spatial rotations return to the identity after more than one cycle (Fig. 3b), denoted $\hat{C}_{n,m}$. This type of DS is exhibited by bi-circular EM fields of frequencies $m\omega$ and $(n-m)\omega$[32,42], and is expressed as

$$\hat{C}_{n,m} = \hat{\tau}_n \cdot \left(\hat{R}_n\right)^m \equiv \hat{\tau}_n \cdot \hat{R}_{n,m} \qquad (8)$$

For $m=1$ this operator reduces to Eq. (7).

The above DSs contain spatial transformations that are all symmetries in molecular groups[2]. We have also discovered a type of DS with a spatial term that has no analog in molecules. This is a discrete elliptical symmetry that generalizes Eq. (8) by considering rotations along an ellipse instead of a circle. By convention, we define elliptical symmetries to always have their major axis along the $x$-axis. This symmetry is expressed as products of rotation and scaling operators:

$$\hat{e}_{n,m} = \hat{\tau}_n \cdot \hat{L}_b \cdot \hat{R}_{n,m} \cdot \hat{L}_{1/b}, \text{ where} \qquad (9)$$

$$\hat{L}_b = \begin{pmatrix} 1 & \\ & b \end{pmatrix} \qquad (10)$$

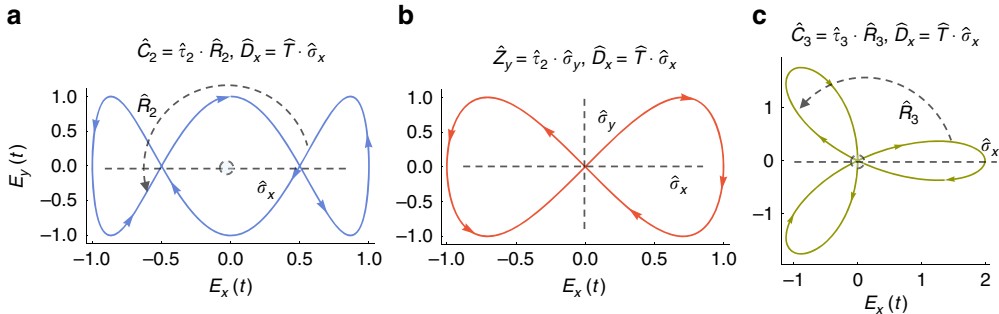

**Fig. 4** Example time-periodic (2+1)D fields characterized by groups with two generators. **a** A group with the generators $\hat{C}_2$ and $\hat{D}_x$ for the example field $\mathbf{E}(t)$ $=\sin(\omega t)\hat{x}+\cos(3\omega t)\hat{y}$, (**b**) a group with the generators $\hat{Z}_y$ and $\hat{D}_x$ for the example field $\mathbf{E}(t)=\sin(\omega t)\hat{x}+\sin(2\omega t)\hat{y}$, and (**c**) a group with the generators $\hat{C}_3$ and $\hat{D}_x$ for the example field $\mathbf{E}(t)=(\cos(\omega t)+\cos(2\omega t))\hat{x}+(\sin(\omega t)-\sin(2\omega t))\hat{y}$. The fields are represented on Lissajou plots. The spatial parts of the operators are indicated by dashed lines and arrows, colored arrows in the plots indicate the direction of time

is a scaling transformation along the elliptical axis spanned in Cartesian coordinates, and $0 \leq b \leq 1$ is the ellipticity of the underlying symmetry (see example third-order case in Fig. 3c).

The DSs described above form the basis of dynamical groups (or Floquet groups). Physically, this means that a specific Floquet system can simultaneously exhibit several types of DSs, that is, its symmetry properties are uniquely defined by a set of generating operators. We mention several examples (more examples are given in supplementary note (SN) 4 in supplementary table 1, see SN 5 for further discussion). First, there are order-2 groups with one generator (examples seen in Figs. 1 and 2), and cyclic groups of higher order that have $\hat{C}_{n,m}$ or $\hat{e}_{n,m}$ as generators (Fig. 3a), which are all abelian. Groups with two generators can be either abelian (Fig. 4a, b), or non-abelian (Fig. 4c).

**DSs and groups in (3+1)D.** We now consider the full (3+1)D case. In terms of the symmetry operations themselves, this means upgrading reflection axes to planes, and rotational operators to revolve around specific axes. Thus, all DSs in Eqs. (1)–(10) are extended to the 3D case by choice of plane or axis. For example, elliptical rotations are extended to 3D by denoting the $z$-axis as the rotation axis, and specifying the axis around which the scaling transformation occurs (within the $xy$ plane):

$$\hat{e}_{n,m} = \hat{\tau}_n \cdot \hat{L}_b^y \cdot \hat{R}_{n,m}^z \cdot \hat{L}_{1/b}^y, \text{ where} \tag{11}$$

$$\hat{L}_b^y = \begin{pmatrix} 1 & & \\ & b & \\ & & 1 \end{pmatrix} \tag{12}$$

An example for the (3+1)D case of the $\hat{C}_2$ DS is shown in Fig. 5a. Beyond these DSs, (3+1)D also offers DSs that have no analog in (2+1)D. These include order-2 DSs with spatial inversion appended to temporal operations (see Fig. 5b):

$$\hat{J} = \hat{T} \cdot \hat{i} \tag{13}$$

$$\hat{F} = \hat{\tau}_2 \cdot \hat{i} \tag{14}$$

$$\hat{A} = \hat{T} \cdot \hat{\tau}_2 \cdot \hat{i} \tag{15}$$

and DSs that involve spatial improper rotations, which are products of a rotation and a reflection about the plane normal to the rotation axis. There are two different improper rotational DSs, for even or odd orders, respectively. For even orders we define the

DS:

$$\hat{M}_{2n,m} = \hat{\tau}_{2n} \cdot \hat{\sigma}_h \cdot \hat{R}_{2n,m} \equiv \hat{\tau}_{2n} \cdot \hat{s}_{2n,m}, \tag{16}$$

whereas for odd orders we have:

$$\hat{M}_{2n+1,m} = \hat{\tau}_{2(2n+1)} \cdot \hat{s}_{2n+1,m} \tag{17}$$

where $\hat{M}_{2n,m}$ is order $2n$, $\hat{M}_{2n+1,m}$ is order $2(2n+1)$, and the index $m$ is equivalent to that in Eq. (8) (for example see Fig. 5c). The operator in Eq. (16) was previously considered in cross-beam geometries[20]. Naturally, these symmetries can be generalized to an elliptical case by replacing the standard circular rotations with elliptical ones. This leads to improper elliptical rotation DSs:

$$\hat{P}_{2n,m} = \hat{\tau}_{2n} \cdot \hat{\sigma}_h \cdot \left(\hat{L}_b \cdot \hat{R}_{2n,m} \cdot \hat{L}_{1/b}\right) = \hat{e}_{2n,m} \cdot \hat{\sigma}_h \tag{18}$$

$$\hat{P}_{2n+1,m} = \hat{\tau}_{2(2n+1)} \cdot \hat{\sigma}_h \cdot \left(\hat{L}_b \cdot \hat{R}_{2n+1,m} \cdot \hat{L}_{1/b}\right) \tag{19}$$

All of the (3+1)D DSs construct similar groups to the (2+1)D case that have three finite spatial dimensions, and an additional infinite and periodic time axis. For example, the field shown in Fig. 5c possesses not only improper rotational DS of order 4, but also a two-fold rotational DS around the $z$-axis ($\hat{C}_2$), as well as other DSs which all together form a dynamical group.

**Dynamical symmetry selection rules.** We derive the physical constraints that arise in systems exhibiting DSs. We consider a general periodic time-dependent Hamiltonian: $H(t) = H(t + T)$. The Floquet Hamiltonian is defined as[43]

$$\mathcal{H}_F = (H(t) - i\partial_t), \tag{20}$$

whose eigenstates are $T$-periodic Floquet modes ($|u_n(t)\rangle$), with corresponding quasi-energies ($\varepsilon_n$). Solutions to the time-dependent Schrödinger equation (TDSE) are comprised of Floquet states: $|\phi_n(t)\rangle = \exp(i\varepsilon_n t)|u_n(t)\rangle$, which we assume are nondegenerate. If $H(t)$ conforms to a DS group $G$, then the Floquet Hamiltonian commutes with all operators in $G$:

$$[\hat{X}, \mathcal{H}_F] = 0, \hat{X} \in G \tag{21}$$

In this case, the Floquet modes are simultaneous eigenmodes of the Floquet Hamiltonian, and of $\hat{X}$. Also, since $\hat{X}$ is unitary or anti-unitary, its eigenvalues are roots of unity:

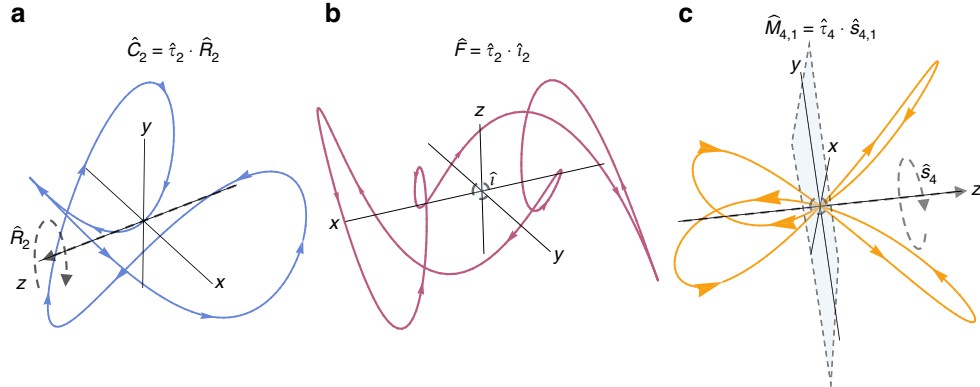

**Fig. 5** Example time-periodic fields exhibiting (3+1)D DSs. **a** $\hat{C}_2$ symmetry for the example field $\mathbf{E}(t) = \sin(3\omega t)\hat{x} + \cos(\omega t)\hat{y} + \cos(2\omega t)\hat{z}$, (**b**) $\hat{F}$ symmetry for the example field $\mathbf{E}(t) = \cos(\omega t)\hat{x} + \sin(3\omega t)\hat{y} + \sin(5\omega t + \pi/6)\hat{z}$, and (**c**) $\hat{M}_{4,1}$ symmetry for the example field $\mathbf{E}(t) = (\cos(\omega t) + \cos(3\omega t))\hat{x} + (\sin(\omega t) - \sin(3\omega t))\hat{y} + \cos(2\omega t)\hat{z}$. The fields are represented on 3D Lissajou plots. The spatial parts of the operators are indicated by dashed arrows and planes, colored arrows in the plots indicate the direction of time

$\hat{X}|u_n(t)\rangle = e^{i\theta_n}|u_n(t)\rangle$, where $\theta_n$ is real. Thus, if the initial wave function populates a single Floquet state, the symmetry directly manifests in the time-dependent probability density $||\phi_n(t)\rangle|^2$ (or any other quantity that does not depend on the phase of the state). In this case, any measured observable **o**(t) also upholds the DS:

$$\mathbf{o}(t) = \langle u_n(t)|\hat{\mathbf{O}}|u_n(t)\rangle$$
$$= \langle u_n(t)|\hat{X}^\dagger \cdot \hat{X} \cdot \hat{\mathbf{O}} \cdot \hat{X}^\dagger \cdot \hat{X}|u_n(t)\rangle = \hat{X} \cdot \mathbf{o}(t) \quad (22)$$

where $\hat{X}^\dagger$ is the inverse of $\hat{X}$. Any observable then complies to all the spatiotemporal symmetries in the group $G$, which imposes selection rules on its spectrum (Eq. (22) can be generalized to spatially dependent observables such as flux operators by transforming the spatially dependent part). In SN 1 we show that Eq. (22) can be re-written as a set of eigenvalue problems in the Fourier domain. Also, in SN 2 we show that the conditions imposed on the temporal evolution can be expressed by the generating DSs in $G$ alone. If $G$ is non-abelian, $|u_n(t)\rangle$ can become degenerate, in which case Eq. (22) applies only to a set of commuting operators in $G$, of which the initial state is chosen as an eigenmode. Still, for an isotropic ensemble (that contains an equal population of all degenerate states) the observables are symmetric under all operations in $G$ even if it is non-abelian, since the sum of the projections on all degenerate states is always symmetric. $|u_n(t)\rangle$ may also become degenerate if $G$ contains time-reversal symmetry and the system has a non-integer spin (through Kramers degeneracy[44]). This condition results from the anti-unitarity of time reversal, as opposed to spatial reflections which are unitary, hence this feature does not have an analog in previously derived line-groups[41] and space–time groups[39]. In this case, Eq. (22) still holds for isotropic ensembles that populate an equal amount of the Kramers pairs (the degenerate time reversal partner states) due to the same conditions described above. Lastly, we note that in a homogenous chiral ensemble, any DSs that involve spatial reflections, inversions, or improper rotations will not lead to selection rules as a consequence of the medium breaking those symmetries[45].

**Selection rules for HG.** As an example, we apply the theory to HG, deriving the selection rules for important symmetries, and discussing several symmetry breaking mechanisms.

We begin by analyzing the symmetries of the HG Hamiltonian. Within the Born–Oppenheimer (BO) and dipole approximations,

the microscopic Hamiltonian of a nonlinear medium interacting with a laser field is given in atomic units and in the length gauge by

$$H_{\mathrm{HG}}(t) = -\frac{1}{2}\sum_j \nabla_j^2 + \frac{1}{2}\sum_{i \neq j}\frac{1}{|\mathbf{r}_i - \mathbf{r}_j|}$$
$$+ \sum_j V(\mathbf{r}_j) + \sum_j \mathbf{E}(t) \cdot \mathbf{r}_j \quad (23)$$

where $H_{\mathrm{HG}}$ is the full time-dependent multi-electron Hamiltonian, $\mathbf{r}_j$ is the coordinate of the $j$'th electron, $\nabla_j^2$ is the Laplacian operator with respect to $\mathbf{r}_j$, $V(\mathbf{r})$ is the potential energy term, $\mathbf{E}(t)$ is the electric driving laser field, and we neglect spin–orbit interactions. This Hamiltonian describes HG in most atomic, molecular, and solid media (the theory can in principle be applied to any Hamiltonian, for example, Hamiltonians that include spin–orbit terms, non-dipole light–matter interaction terms, etc.). For a DS to result in selection rules, it should commute with the Hamiltonian, that is, commute with all four terms in Eq. (23). Notably, the first three terms in $H_{\mathrm{HG}}$ are time-independent, and therefore invariant under any temporal operation. Furthermore, the kinetic and electron–electron interaction terms are both invariant under all possible rotations and reflections. The spatial symmetries of $V(\mathbf{r})$ can be analyzed by standard group theory. It is then left to match the symmetries of $V(\mathbf{r})$ with the spatial part of the DSs. The remaining laser–matter interaction term is the only time-dependent term in $H_{\mathrm{HG}}$, and the analysis of its symmetries is equivalent to analyzing the DSs of $\mathbf{E}(t)$. Overall, for a DS to commute with $H_{\mathrm{HG}}$, the driver field should exhibit the DS, and the potential term should exhibit the spatial part of the DS.

In HG, the high harmonic spectrum is found by Fourier transforming the time-dependent polarization:

$$\mathbf{p}(t) = \langle\psi(t)|\hat{\mathbf{r}}|\psi(t)\rangle \quad (24)$$

where $\psi(t)$ is the solution to the TDSE. DSs that commute with the Hamiltonian impose constraints on $\mathbf{p}(t)$ (Eq. (22)), and lead to selection rules on the harmonic spectrum. To derive the selection rules, we analyze a general $\mathbf{p}(t)$ function that upholds the constraints in the Fourier-domain (SN 3).

Table 1 presents selection rules for (2+1)D DSs in collinear HG. These include four selection rules: (1) a reflection time-reversal symmetry ($\hat{D}$, $\hat{H}$) that results in elliptically polarized harmonics, where the elliptical major or minor axis corresponds to the reflection axis. (2) When HG is driven by a laser field

**Table 1 (2+1)D DSs and their associated selection rules for collinear atomic/molecular HG**

| Symmetry | Order | Harmonic generation selection rule |
|---|---|---|
| $\hat{D}, \hat{H}$ | 2 | Elliptically polarized harmonics with major/minor axis corresponding to the reflection axis. |
| $\hat{T}, \hat{Q}, \hat{G}$ | 2 | Linearly polarized only harmonics. |
| $\hat{Z}$ | 2 | Linearly polarized only harmonics, even harmonics are polarized along the reflection axis, and odd harmonics are polarized orthogonal to the reflection axis. |
| $\hat{C}_2$ | 2 | Odd-only harmonics, any polarization is possible. |
| $\hat{C}_{n,m}$ | $n > 2$ | ($\pm$) circularly polarized ($nq \mp m$) harmonics, $q \in \mathbb{N}$, all other orders forbidden. |
| $\hat{e}_{n,m}$ | $n > 2$ | ($\pm$) elliptically polarized ($nq \mp m$) harmonics, $q \in \mathbb{N}$, with an ellipticity $b$, all other orders forbidden. |

**Table 2 (3+1)D DSs and their associated selection rules for collinear or non-collinear atomic/molecular/solid HG**

| Symmetry | Order | Harmonic generation selection rule |
|---|---|---|
| $\hat{D}, \hat{H}$ | 2 | The polarization ellipsoid has a major/minor axis normal to the reflection plane. |
| $\hat{Q}, \hat{G}$ | 2 | The rotation axis is a major/minor axis of the polarization ellipsoid. |
| $\hat{Z}$ | 2 | Odd harmonics are polarized linearly and orthogonally to the reflection plane, only even harmonics allowed polarized within the reflection plane |
| $\hat{C}_2$ | 2 | Odd-only harmonics in any polarization are allowed polarized in the plane orthogonal to rotation axis, even only harmonic emission is allowed polarized parallel to the rotation axis. |
| $\hat{T}, \hat{J}, \hat{A}$ | 2 | Linearly polarized harmonics only. |
| $\hat{F}$ | 2 | Odd-only harmonics in any polarization. |
| $\hat{e}_{n,m}$ | $n > 2$ | ($\pm$) elliptically polarized ($nq \mp m$) harmonics, $q \in \mathbb{N}$, with an ellipticity $b$ within the plane orthogonal to rotation axis. Linearly polarized $nq$ harmonics are also allowed, but polarized parallel to the rotation axis. |
| $\hat{P}_{2n,m}$ | $2n > 2$ | ($\pm$) elliptically polarized ($2nq \mp m$) harmonics, $q \in \mathbb{N}$, with an ellipticity $b$ within the plane orthogonal to the improper rotation axis. $n(2q+1)$ harmonics are also allowed, but polarized parallel to improper rotation axis. |
| $\hat{P}_{2n+1,m}$ | $2(2n+1) > 2$ | ($\pm$) elliptically polarized $2q(2n+1) \mp 2m$ harmonics, $q \in \mathbb{N}$, with an ellipticity $b$ within the plane orthogonal to the improper rotation axis. $(2n+1)(2q+1)$ harmonics are also allowed, but polarized parallel to improper rotation axis. |

which is time-reversal ($\hat{T}$) invariant, the harmonics are linearly polarized. Notably, such fields (see supplementary Figs. 3, 4) can lead to rich two-dimensional electron dynamics and re-collisions from multiple directions. Nonetheless, symmetry dictates that the different contributions interfere in a manner that leads to linearly polarized harmonics only. (3) A reflection translation symmetry ($\hat{Z}$) results in even harmonics that are linearly polarized along the symmetry axis, and odd harmonics that are linearly polarized perpendicular to it, as recently demonstrated experimentally by HHG from cross-linear $\omega - 2\omega$ pumps[45,46]. Our results indicate that this geometry is a subset of a wider array of laser fields that uphold this symmetry (examples are shown in supplementary Figs. 1, 2). (4) Elliptical symmetry ($\hat{e}_{n,m}$) results in a high harmonic spectrum where all harmonic orders have exactly the same ellipticity, which corresponds to the ellipticity parameter, $b$, of the Hamiltonian's underlying symmetry. The helicities alternate between harmonic orders, similar to the circular case ($\hat{C}_{n,m}$). Elliptically polarized high harmonics with fully tunable ellipticity may be useful for ultrafast spectroscopy and HG-based ellipsometry[47]. Notably, the fact that the ellipticity is determined by the DS makes it much more robust to perturbations than in other techniques[30].

For the (3+1)D case that describes non-collinear geometries, DSs and their associated selection rules are presented in Table 2. We note two selection rules: (1) $n$-fold rotational DS ($\hat{e}_{n,m}$) leads to $nq$ harmonics that are all allowed (for integer $q$), suffice they are polarized only along the rotation axis. This can lead to photon mixing between all three polarization axes. (2) Improper rotational DSs ($\hat{P}_{n,m}$) lead to selection rules with symmetry-forbidden harmonics, as well as elliptically/circularly polarized harmonics. These are both numerically verified and discussed in SN 6 (see supplementary Figs. 7, 8).

The symmetry of the system (the Hamiltonian in Eq. (23)) is determined by the DSs of the pump field, and the symmetries of

the potential $V(\mathbf{r})$. Often, HG is driven in atomic gas, and $V(\mathbf{r})$ is spherically symmetric; hence, selection rules are a consequence of only the DSs of the pump field (see SN 6). Still, there are various interesting cases when the medium is not spherically symmetric that are worth discussing. First, randomly distributed (non-oriented) molecular gas. In this case, the pump field interacts with all orientations of the molecule, and it is therefore the symmetry of the orientation averaged ensemble that affects the HG selection rules. The orientation-averaged ensemble is either $O(3)$ symmetric if it is achiral, or $SO(3)$ symmetric if it is chiral. The $O(3)$ group is spherically symmetric, so HG from non-oriented achiral media leads to the same selection rules as those observed from atomic spherically symmetric media[48]. Notably, the symmetries of the molecule are washed-out, and do not lead to selection rules in the HG spectrum (see SN 7 and supplementary Fig. 9). This picture is also valid for randomly distributed chiral media, except that $SO(3)$ does not contain reflection symmetries, so reflectional-based selection rules are broken (which can be used for chiral spectroscopy[49]). Second, a gas of aligned or oriented molecules[38,48,50–52]. Here, $V(\mathbf{r})$ depends on the relative orientation with respect to the laser field, and selection rules are only observed if both the medium and pump are co-aligned along the DS axis/plane (see SN 7 and supplementary Fig. 10). Third, HG in solid media[53–56]. Some selection rules were previously derived for HG in solids, but these either dealt with rotational DSs[57], or considered rotation/inversion symmetries of the solid while assuming it interacts with a monochromatic pump[58,59]. Here we derived the general case that accounts for the DSs of the incident laser field, as well as those of the solid, which requires the (3+1)D formulation given in Table 2 even for collinear cases (since the crystal structure can couple different spatial axes). Thus, apart from the $\hat{C}_n$ operator, the DSs and selection rules presented in Table 2 are new in solid HG (for details see SN 8). Here the symmetries of the solid and the respective angle of

incidence of the pump can determine the system's overall symmetry, which could be used for HG spectroscopy. For example, DS breaking spectroscopy can be used for orientation spectroscopy (see supplementary Fig. 11), as well as to probe spin–orbit and magnetic interactions (which break reflectional-based DSs and time-reversal-based DSs, respectively).

The analysis above assumes that the Hamiltonian is fully invariant under the DS transformations. In reality, HG Hamiltonians are not perfectly invariant. For instance, ionization and the driver pulse finite duration perturb time-reversal and translation symmetries. In fact, time-reversal symmetry can be significantly broken when the ionization rate per optical cycle is large, because field-induced tunneling is a non-reversible process (this effect is significant for harmonics above the ionization potential—see SN 6). Still, selection rules in HG are observed routinely, both numerically and experimentally[19,30,48]. The actuality of the selection rule associated with the elliptical symmetry operation, $\hat{e}_{n,m}$, is more subtle, because the kinetic energy and potential terms are generally not invariant under the elliptical transformation. One approach to address this inconsistency is to use drivers that exhibit both spherical and elliptical DSs. A burst of linearly polarized pump trains[60,61] and a synthetic piecewise driver pump can conform to this condition (see SN 6 and supplementary Figs. 5, 6). Another approach is to generate harmonics in media with an elliptically symmetric kinetic term, for instance by utilizing the concept of transformation optics[62]. It is also worth mentioning that propagation effects may influence the selection rules of macroscopic systems. For example, individual harmonic polarizations or intensities may change during propagation (but not forbidden harmonic selection rules). In fact, specially designed configurations may lead to generation of circularly polarized high harmonics, even though the emission from each atom is polarized linearly[63], or to quasi-phase matching of only the even-order harmonics[64]. Notably, the presented theory can be extended to include propagation effects by applying the DS group theory approach also to Maxwell's equations and give rise to new selection rules in HG.

Lastly, we discuss the link between DSs and selection rules to conservation laws. Although Noether's theorem does not connect discrete DSs to conservation laws, all previous selection rules in HG were also derived from conservation laws: the appearance of only discrete harmonics in the spectrum (the selection rule due to time-periodicity $\hat{\tau}_1$) can be derived from energy conservation, the $\hat{C}_2$ selection rule can be derived from parity conservation, and the $\hat{C}_n$ selection rule can be derived from conservation of spin angular momentum of the interacting photons[65,66]. This duality is believed to reflect an equivalence between the DS and photonic pictures in HG[65]. Interestingly, we find that several of the DSs ($\hat{e}_{n,m}$, $\hat{P}_{n,m}$, $\hat{Z}$, $\hat{T}$, $\hat{Q}$, $\hat{D}$, and $\hat{J}$) lead to selection rules that have no analog conservation law derivation. What does this result mean? It could indicate that the DS perspective is more general than the photonic perspective. Can conservation laws associated with the above DSs save the disparity between the two approaches?

## Discussion

In this paper, we have implemented the concepts of group theory to map and characterize DSs exhibited by Floquet systems. We introduced symmetries that involve time-reversal operations, and an elliptical symmetry that does not exist in molecular groups. We proved that if a given Hamiltonian commutes with a dynamical group then the generators of the group constrain the temporal evolution of Floquet states and the associated physical observables. We described these symmetry constraints as eigenvalue problems in the Fourier domain, and derived their resulting selection rules.

For collinear atomic and molecular HG, we discovered several selection rules: a reflection-time-reversal DS ($\hat{D},\hat{H}$) results in elliptically polarized harmonics with an elliptical major/minor axis parallel to the reflection axis. Any driver with time-reversal ($\hat{T}$), or time-reversal followed by $\pi$ rotations ($\hat{Q},\hat{G}$) results in linearly polarized harmonics. A spatial reflection followed by temporal translation ($\hat{Z}$) results in linearly polarized harmonics, where even harmonics are polarized along the symmetry axis, and odd harmonics are orthogonal to it[45]. An elliptical DS ($\hat{e}_{n,m}$) results in elliptically polarized eigenstates, and is useful for robust control of the ellipticity of all the high-order harmonics, collectively. These high harmonics would be useful for HG-based ellipsometry[47]. Furthermore, we explored non-collinear and solid HG and found that these include DSs that do not exist in the (2 +1)D case: an inversion-time-translation DS that leads to odd-only harmonics ($\hat{F}$), an inversion-time-reversal DS that leads to linearly polarized harmonics ($\hat{J},\hat{A}$), and improper-rotational DSs that lead to circularly/elliptically polarized high harmonics within the symmetry plane, and linearly polarized harmonics orthogonal to it ($\hat{M}/\hat{P}$). Interestingly, in contrast to current selection rules that can be derived by DSs or equivalently from conservation laws, some of the selection rules cannot be derived by the known conservation laws.

A very exciting aspect of the HG group theory is that it forms a starting point for utilizing and investigating broken symmetries. In this respect, even if a DS leads to a 'boring' selection rule (for instance, time-reversal DS, $\hat{T}$, that 'just' leads to a linearly polarized spectrum), it still has major implications for ultrafast spectroscopy of various systems. In this sense, there are no boring selection rules. For example, the HG group theory can be used to characterize the symmetries and determine the orientation of any molecular or solid medium[48] (see SN 7, 8), time-translation and time-reversal DSs ($\hat{\tau}_n$, $\hat{T}$) can be used to find and characterize atomic and molecular resonances and probe ionization dynamics (see SN 6), time-reversal DSs ($\hat{T}$, $\hat{Q}$, $\hat{D}$, $\hat{J}$) can be used to probe magnetic interactions, reflection DSs ($\hat{Z}$, $\hat{D}$) can shed light on spin–orbit interactions in atomic, molecular, and solid systems, and as will be reported soon[49], reflection/inversion DSs ($\hat{Z}$, $\hat{D}$, $\hat{J}$, $\hat{F}$) can be used to probe dynamical chiral processes[67], and more.

We applied here the derived Floquet group theory for analyzing HG at the microscopic level. Its application for exploring wave propagation effects will follow soon. Our work also paves the way for several interesting directions beyond nonlinear optics. The introduced elliptical DS can be implemented in other systems, both static (such as metamaterials[22,23] and transformation optics[62]) and Floquet, to yield elliptical eigenstates. Extending the theory to lattice systems by including translational operators gives rise to dynamical space groups, which may lead to group theory-based classification of Floquet topological insulators[3,6], time-crystals[68], and shaken optical lattices[9,10]. Extensions to non-Hermitian systems including $PT$ symmetric waveguides[69], and quasi-periodic Floquet systems[70] should also be possible and exciting, leading to new DSs and selection rules. Overall, we expect that group theory analysis of DSs will lead to extended understanding and novel discoveries in various Floquet systems.

## Data availability

All relevant data are included in the main manuscript and the Supplementary information. Additional data are available from the corresponding authors upon reasonable request.

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

## Acknowledgements

This work was supported by the Israel Science Foundation (grant no. 1781/18), the Israeli Center of Research Excellence "Circle of Light" supported by the I-CORE Program of the Planning and Budgeting Committee and the Israel Science Foundation (grant no. 1802/12), the joint UGC-ISF Research Grant Program (grant no. 1903/14), and the Wolfson foundation. D.P. thanks support from the National Science Foundation through a grant to ITAMP at the Harvard-Smithsonian Center for Astrophysics. O.N. gratefully acknowledges the support of the Adams Fellowship Program of the Israel Academy of Sciences and Humanities.

## Author contributions

O.N. and O.C. initiated this research direction. O.N. did everything presented in this paper, D.P. assisted with the formal theory and O.C. supervised the project. All authors wrote the paper.

## Additional information

**Competing interests:** The authors declare no competing interests.

