## [Peer Review File · Nature Communications]

Editorial Note: This manuscript has been previously reviewed at another journal that is not operating a transparent peer review scheme. This document only contains reviewer comments and rebuttal letters for versions considered at Nature Communications. Mentions of prior referee reports have been redacted.

Reviewers' comments:

Reviewer #1 (Remarks to the Author):

In my last two review reports, I already confirmed the significance of this work, which is timely and important and will be of general interest for researchers in different areas. In my first report for the first edition of this paper, I suggested the authors to enhance the originality. In the second edition, the authors had extended their consideration to more general cases and make their study more systematically. Some of the discussed topics, like the HHG of chiral molecules and solids and their spectral applications, are the forefronts of the strong-field research area. Therefore, I think the originality had been enhanced. However, I felt some of the important points were still not clear for the readers in the second edition. So I suggested the authors to add more detailed information and to further clarify the discussions.

In this edition, all the questions are well addressed. I think the current paper is of significance and novelty and is complete now. I therefore recommend publication of this paper in Nature Communications.

Some of the references, mostly the APS papers, are still not correctly cited.

Reviewer #2 (Remarks to the Author):

[Redacted] I would particularly like to thank the authors for adding the clear discussion of the connexion between the DS groups and the selection rules - which in turn opens several exciting research directions. Otherwise, I feel that my assessment of the significance and novelty of this work remains unchanged.

I believe that this work will be a very influential and timely addition to the field on non-linear laser-matter interactions. Personally, I find that it already strongly influences the way I think about high-harmonics generation. I am looking forward to see it published in Nature Communications.

Two cosmetic comments:

1. On line 242, please replace "suffice they" by "which"
2. On lines 401-402, ref. 29 is incomplete - it misses one of the authors' names and the journal title.

My overall recommendation is to publish without change.

Reviewer #3 (Remarks to the Author):

[Redacted] the authors have not managed to improve neither novelty nor generality of this work.

Symmetry is of course a standard approach in science to identify conservation laws and simplify the analysis. If one finds or postulates certain symmetries for the Hamiltonian (and time evolution), it is common wisdom that the system response will follow them. This kind of symmetry identification has already been performed for harmonic generation (HG). The presented work is clearly more complete than the previous efforts, but presenting a clean up of existing approaches does not yet count as novel result.

Most of HG symmetry investigations have been conducted for atomic/molecular systems, so the novelty could arise from new insights provided for solid-state HG. Unfortunately, the presented results cannot be applied to solids for multiple reason. For solids, the starting-point Eq. (23) contains an unknown potential $V(r)$ with unknown symmetries (known only by solving the full many-body problem, which nobody has managed to do). As long as $V(r)$ is unknown, one can only postulate approximate symmetries to be tested. One possibility is to use tested pseudopotentials or DFT results. Such an approximated $V(r)$ will satisfy, e.g., invariance with respect to certain translations and spin-operations; most common way is to analyze them via band structure (not attempted in the present work). The authors have omitted both aspects from the analysis is not applicable for solids. Even when they were included, there is a next level of relevant, but omitted effects – phonons that inherently distort the symmetries related to $V(r)$. The phonon effects cannot be included via Eq. (23) despite they are important for optical properties of solids. As a further remark, Eq. (23) cannot be solved using the time-dependent Schrödinger equation (method used in this work) for more than 3 electrons, which is simply insufficient to draw any conclusions how solids work (were $V(r)$ and phonon effects somehow included). In summary, this work is an oversimplification for solids and does not really provide new insights how solid-state HG works.

At the same time, this paper makes interesting advancement for atomic/molecular HG. This work will definitely be interesting for experts while it lacks general novelty and broad scientific impact. Therefore, a more specialized journal would be more appropriate after the overstatements for the solid-state implications are toned down. [Redacted]

Reviewer #4 (Remarks to the Author):

The authors present a novel approach for generating symmetry relationships through consideration of commutators of the time-dependent Hamiltonian in optical harmonic generation. This architecture is distinct from widely used descriptions based on time-dependent perturbation theory, with possible advantages with respect to extension to high harmonic generation (in which perturbation theory becomes unwieldy). The authors target applications for describing polarization-dependent high harmonic generation in gasses, surfaces, and bulk media.

1. The formulation as developed appears to have been applied to only a small number of different assumed symmetries for the ensembles. The most common case of isotropic assemblies, consistent with gasses and liquids makes sense and is the most straightforward (and most well-described elsewhere). The more interesting cases in which the proposed approach may offer the greatest benefits are for bulk assemblies of lower symmetry. The authors correctly state that the approach they have developed also applies to such assemblies, but work through only a small handful of specific examples (e.g., SHG from an oriented gas ensemble of D_{3h} symmetry, and an assembly of Z symmetry, corresponding to a mirror plane and a temporal translation). There are 32 crystal classes corresponding to different point group symmetries. It is not clear how the proposed theoretical framework can be used to predict the polarization-dependence of high harmonic generation for these specific cases.

2. Stronger connections between the proposed time-dependent Hamiltonian approach and conventional polarization-dependent nonlinear optics in cases in which they coincide would be helpful in identifying the reasoning behind the conditions leading to divergence.

3. The utility of the time-dependent approach taken in this work is difficult to assess for interpretation of high harmonic generation. While harmonic generation in isotropic assemblies is reasonably straightforward, the enabling capability of the proposed approach is arguably the extension to assemblies of lower symmetry. Is this easy to do for a high harmonic from an assembly of low symmetry? How are the results to be interpreted?

4. This work describes a fairly specialized topic of polarization-dependent high harmonic generation from a purely theoretical perspective. As such, the work is likely to appeal to a relatively small fraction of the Nature Communications readership. It may be more appropriate to consider an alternative topical journal for publication, in which the target audience is more likely to come across the work.

To referees:

Before replying to the referee's comments in the last round of review, we kindly wish to emphasize to all reviewers that [Redacted] the following questions that are now critical for the editorial decision are: **1) Is our theory of symmetries and selection rules in Floquet systems general and specifically applicable to harmonic generation from bulk crystals? 2) Would it have significant impact on high harmonics from bulk samples?** Our answer to the above two questions is an emphatic yes.

First, it may be instructive to elucidate the **generality and applicability of our theory to harmonic generation from bulk crystals** less formally than in the manuscript:

1. We remind that in perturbative non-linear optics in bulk crystals, the medium's symmetries (i.e. the crystallographic structure of the crystal) dictate selection rules on the emitted spectra. In the perturbative approach, this is often developed by deriving symmetry constraints on the non-linear optical coefficient tensors of the material (see for example section 1.5 in Boyd's book "Nonlinear Optics"). A standard textbook example is the absence of SHG in centrosymmetric media.
2. We also remind that several dynamical symmetries (DSs) of the pump lasers are known to lead to selection rules in HHG spectra from atomic gas and isotropic media. For example, ω - 2ω bi-circular pumps lead to $3q+1$ circularly-polarized harmonics with helicity like the ω component of the pump, $3q+2$ circularly-polarized harmonics with helicity like the 2ω component of the pump, and $3q$ forbidden harmonics (refs. 26, 36 in the paper).
3. Our theory – a group theory derivation for selection rules in Floquet systems and its application to harmonic generation – takes into account both the DSs of the pump and the symmetry of the medium, and is therefore more general than the theories and models of points (1) and (2). That is, our theory covers ALL the selection rules in (1) and (2), as well as many more. Indeed, we uncovered novel and nonintuitive symmetries and their associated selection rules that have never been considered (e.g. elliptical, reflectional, time-reversal, non-collinear geometries, etc.).
4. In practice, the combination of the crystal/molecule symmetry and pump symmetry is done by taking the following steps: i) identify the symmetry of the medium (molecules or crystals) using standard well-known and fundamental tools (molecular group theory for molecules, and crystallographic space group theory for solids), as is standard in perturbative nonlinear optics. ii) Identify the DS of the pump (according to the first section in the paper, and the SI). iii) Identify the DS of the joint system (pump and medium) through dynamical group theory analysis (according to the last section in the paper, and the SI). (iv) Find the selection rules corresponding to these DSs in the tables presented in our paper. Of course, our paper gives the general mathematical formalism for this procedure. Notably, the selection rules are derived analytically based only on the fact that the system (Hamiltonian) exhibits the joint DSs.
5. In case one argues that HHG in bulk crystals includes something that prevents the observability of selection rules, we wish to remind that such selection rules were already observed experimentally (refs. 31,60,62 in the paper). Of course, our theory agrees with all these observed selection rules.

In summary, we stand firm that our derived theoretical framework (resulting in tables I and II in the paper), covers ALL previous selection rules of harmonic generation from crystals that are not due to propagation effects, and introduces NEW selection rules (including reflection, inversion, elliptical, time-reversal, and non-collinear DSs, applicable also to harmonic generation from bulk crystals). We challenge reviewers that oppose our claim to specify a contradicting example.

Impact specifically on harmonic generation from bulk crystals:

- We derive many new symmetries for (2+1)D and (3+1)D cases leading to new selection rules, including: i) an elliptical DS, ii) reflectional DS, iii) a variety of symmetries that involve time-reversal coupled to spatial symmetries, iv) DSs that exist only in (3+1)D such as spatial inversion and improper rotations coupled to various temporal operations. The new symmetries will lead to enhanced control of EUV radiation emitted from bulk crystals. Moreover, because the symmetries are analytically derived for HHG from crystals including electron-electron correlations, they could provide new insight into theoretical models for solid HHG.
- In this work we cover the selection rules due to symmetries of the unit cells of crystals. But, the theoretical platform is general and can be extended. We shall soon start preparing a paper that extends the theory to include propagation effects. It also leads to the discovery of more new selection rules. In the future we also plan to apply our theoretical approach to harmonic generation from quasi-crystals.
- The discovery of new symmetries in HG is of immediate importance to ultrafast spectroscopy, where broken symmetries can be investigated to reveal information on the medium or the relevant interactions. In a continuation work we demonstrate this principle for chirality spectroscopy in this recently submitted paper (<https://arxiv.org/abs/1807.02630>). Besides chirality, one can also probe other interactions such as magnetism, spin-orbit coupling, etc., using a similar approach. This will become a highly useful method for ultrafast spectroscopy, including from bulk samples, which may also be used for probing topological properties of matter.

Response to referees #1, 2:

We thank both referees for their hard work and dedication in reviewing our work (now for the fourth time). Thank you! Your comments throughout the process helped us improve the paper into its current form.

Reviewer 1 writes: “In my last two review reports, I already confirmed the significance of this work, which is timely and important and will be of **general interest for researchers in different areas.**”, as well as: “I think the current paper is of significance and novelty and is complete now”.

Referee #2 writes: “I believe that this work will be a **very influential and timely addition to the field on non-linear laser-matter interactions.** Personally, I find that it already strongly influences the way I think about high-harmonics generation.”

We are especially very pleased and grateful for the last sentence in light of the already 30 years and many thousands of papers on HHG.

We hope the referees will give their input on the specific issues that are currently under consideration – application and potential impact of our work in harmonic generation in bulk crystals and the criticism of the other reviewers.

Response to referee # 3 report:

Below we respond point by point to the comments made by the referee, starting with the technical comments that we refute based on basic well-known concepts.

1. With respect to: “For solids, the starting-point Eq. (23) contains an unknown potential $V(r)$ with unknown symmetries (known only by solving the full many-body problem, which nobody has managed to do). As long as $V(r)$ is unknown, one can only postulate approximate symmetries to be tested.”

This comment is blatantly incorrect, and goes against any symmetry theory for solid state physics. The potential $V(r)$, and the Hamiltonian of any solid can be written from first principles, taking the form (in atomic units) [1,2]:

$$H = -\frac{1}{2} \sum_i \vec{v}_i^2 + \frac{1}{2} \sum_{i \neq j} \frac{1}{|\vec{r}_i - \vec{r}_j|} \boxed{\overline{\sum_{i,J} \frac{Z_J}{|\vec{R}_J - \vec{r}_i|}} \equiv V(r)}$$

where \vec{r}_i is the location of the i 'th electron, Z_J the atomic number of the J 'th ion, \vec{R}_J its location, other symbols are as in the paper, and this Hamiltonian already includes the well-known Born-Oppenheimer approximation (J is an ionic summation index and i, j are an electronic summation indices). For simplicity we have omitted spin-orbit interactions (but these can be added without affecting the conclusions). The rightmost term here (in the black box) is exactly the spatial part of the one-electron potential $V(r)$ that the referee thinks is unknown for a solid. This term is exactly the one which is standardly analyzed to catalog a solid according to its symmetry to one of the known space groups [2,3], i.e., the coordination number and geometry of the different ionic species in the crystal indicates the symmetry group regardless of the fact the crystal is infinite. Therefore, the crystal symmetry is decoupled from the solution for the multi-electron wave function of the above Hamiltonian, which is a most basic concept in solid-state physics.

2. With respect to: “One possibility is to use tested pseudopotentials or DFT results. Such an approximated $V(r)$ will satisfy, e.g., invariance with respect to certain translations and spin-operations; most common way is to analyze them via band structure (not attempted in the present work). The authors have omitted both aspects from the analysis is not applicable for solids.”,

These sentences are unrelated to the matter at hand – pseudopotentials are used to approximate the electron-electron interactions from core electrons (the middle term in the above Hamiltonian) [1]. This is irrelevant to the $V(r)$ term which is due to the ionic potential alone, and does not concern electron-electron interactions. DFT is an electronic structure theory, and is again not related to the term $V(r)$, but to solving for the electron density for an already known $V(r)$ [1,4]. The density (and wave function) inherits the symmetries of the potential $V(r)$, because the unitary symmetry operations commute with the Hamiltonian above. This is a well-known fundamental result in QM [5] – one need **not** solve for the wave function in order to catalog the crystal point group [2,3]. Lastly, band theory only represents the symmetries inherited by the real-space potential $V(r)$ above. The k-space and real-space pictures of the Hamiltonian are equivalent by definition, as they are Fourier transforms of one another – if $V(r)$ is reflection invariant, so is the Brillouin-zone (BZ), etc. (the BZ type is in fact also cataloged according to the crystal symmetry, which dictates the high-symmetry lines in k-space). This is often used to reduce computational load, i.e., use the so-called ‘irreducible BZ’ instead of the full BZ. This concept is basic in condensed matter: see (https://en.wikipedia.org/wiki/Crystallographic_point_group, https://en.wikipedia.org/wiki/Brillouin_zone).

3. The referee writes “phonons that inherently distort the symmetries related to $V(r)$. The phonon effects cannot be included via Eq. (23) despite they are important for optical properties of solids”,

Phonons do not affect the symmetry properties of a crystal. This is because, while phonons cause distortions to the lattice, these vibrations occur in different directions as a function of location and time, and therefore the symmetry is restored on average from all macroscopic observables. This is certainly

the case for any optical properties, which self-average in space as the typical beam size is much larger than the lattice constant (and, if the beam duration is longer than typical phonon time scales, then they also self-average in time). This is the reason that crystals are classified according to the 270 space groups, even at non-zero temperatures, despite the presence of thermal phonons.

4. With respect to: “Eq. (23) cannot be solved using the time-dependent Schrödinger equation (method used in this work) for more than 3 electrons, which is simply insufficient to draw any conclusions how solids work”,

There is no need to solve eq. (23) to obtain the symmetry group of the laser-matter system! This is the whole point of symmetry: One can make precise and useful statements without having to solve the Hamiltonian. We only solve this equation in the SI for several examples as to demonstrate that the analytical results hold.

As for the usefulness of the non-interacting electron picture in condensed matter physics (the same approximation we use in the numerical solutions in the SI) - we point out it is the basis for many major results obtained in solid state physics in the last century, including for example the recent Nobel prize for Graphene (whose Dirac cone structure is fully described by a non-interacting electron approximation). Papers on solid HHG nearly all rely on a non-interacting electron picture (just a few references - [6–16]), which leads to results in good agreement with experiments.

Finally, we note that if the reviewer arguments were correct then there wouldn't be selection rules in harmonic generation from bulk crystals, both perturbative and HHG. Fortunately, several selection rules were already observed in several experiments [17–21], two of which actually cited our arXiv paper [20,21]. We argue that ALL selection rules of HHG from bulk crystals (that are not due to propagation effects) are covered by our theory and presented in table II in our paper. We challenge the referee to give a contrary example.

References:

- [1] R. M. Martin, *Electronic Structure: Basic Theory and Practical Methods* (Cambridge university press, 2004).
- [2] C. Kittel, P. McEuen, and P. McEuen, *Introduction to Solid State Physics* (Wiley New York, 1996).
- [3] D. M. Bishop, *Group Theory and Chemistry* (Courier Corporation, 2012).
- [4] W. Koch and M. C. Holthausen, *A Chemist's Guide to Density Functional Theory* (John Wiley & Sons, 2015).
- [5] L D Landau and E.M. Lifshitz, *Quantum-Mechanics (Non-Relativistic Theory)*, 3 edition (Butterworth-Heinemann, 1981).
- [6] S. Ghimire, A. D. Dichiaro, E. Sistrunk, P. Agostini, L. F. Dimauro, and D. A. Reis, *Nat. Phys.* **7**, 138 (2011).
- [7] E. N. Osika, A. Chacón, L. Ortmann, N. Suárez, J. A. Pérez-Hernández, B. Szafran, M. F. Ciappina, F. Sols, A. S. Landsman, and M. Lewenstein, *Phys. Rev. X* **7**, 1 (2017).
- [8] G. Vampa, T. J. Hammond, N. Thire, B. E. Schmidt, F. Legare, C. R. McDonald, T. Brabec, and P. B. Corkum, *Nature* **522**, (2015).
- [9] T. T. Luu and H. J. Wörner, *Phys. Rev. B* **94**, 1 (2016).
- [10] N. Tancogne-Dejean, O. D. Mücke, F. X. Kärtner, and A. Rubio, *Nat. Commun.* **8**, 1 (2017).
- [11] N. Tancogne-Dejean, O. D. Mücke, F. X. Kärtner, and A. Rubio, *Phys. Rev. Lett.* **118**, 1 (2017).
- [12] A. F. Kemper, B. Moritz, J. K. Freericks, and T. P. Devereaux, *New J. Phys.* **15**, (2013).
- [13] Y. S. You, D. A. Reis, and S. Ghimire, *Nat. Phys.* **13**, 345 (2017).
- [14] G. Vampa, C. R. McDonald, G. Orlando, D. D. Klug, P. B. Corkum, and T. Brabec, *Phys. Rev. Lett.* **113**, (2014).
- [15] F. Langer, M. Hohenleutner, U. Huttner, S. W. Koch, M. Kira, and R. Huber, *Nat. Photonics* **11**, 227 (2017).
- [16] M. Wu, D. A. Browne, K. J. Schafer, and M. B. Gaarde, *Phys. Rev. A* **94**, 1 (2016).
- [17] C. L. Tang and H. Rabin, *Phys. Rev. B* **3**, 4025 (1971).
- [18] O. Alon, V. Averbukh, and N. Moiseyev, *Phys. Rev. Lett.* **85**, 5218 (2000).
- [19] A. Yariv and P. Yeh, *Photonics: Optical Electronics in Modern Communications* (Oxford University Press New York, 2007).
- [20] N. Saito, P. Xia, F. Lu, T. Kanai, J. Itatani, and N. Ishii, *Optica* **4**, 1333 (2017).
- [21] N. Klemke, N. Tancogne-Dejean, G. M. Rossi, Y. Yang, R. E. Mainz, G. Di Sciacca, E. Casandru, A. Rubio, F. X. Kärtner, and O. D. Mücke, in *Conf. Lasers Electro-Optics* (Optical Society of America, San Jose, California, 2018), p. FF3P.5.

Response to referee # 4 report:

We thank the referee for reviewing our paper. Below we reply point by point to the referee's remarks:

1. With respect to: “The formulation as developed appears to have been applied to only a small number of different assumed symmetries for the ensembles. The most common case of isotropic assemblies, consistent with gasses and liquids makes sense and is the most straightforward (and most well-described elsewhere). The more interesting cases in which the proposed approach may offer the greatest benefits are for bulk assemblies of lower symmetry. The authors correctly state that the approach they have developed also applies to such assemblies, but work through only a small handful of specific examples (e.g., SHG from an oriented gas ensemble of D3h symmetry, and an assembly of Z symmetry, corresponding to a mirror plane and a temporal translation). There are 32 crystal classes corresponding to different point group symmetries. It is not clear how the proposed theoretical framework can be used to predict the polarization-dependence of high harmonic generation for these specific cases.”
 - First, we thank the reviewer for writing that “The authors correctly state that the approach they have developed also applies to such assemblies.” Thank you. This is important in light of the technical comments of reviewer #3.
 - The theoretical framework for harmonic generation from crystals is used in exactly the same way it is used for the molecular/atomic targets. First, one identifies the symmetry group of the target crystal/molecule and the symmetry operations in the group (for example, is it symmetric under inversion such as MgO, 3-fold rotation such as Al₂O₃, etc.). Second, one identifies the dynamical symmetry group of the laser field (according to criteria given in the paper and SI). Third, any DS operation which is joint to both the target and the laser (i.e., to the full Hamiltonian of the light-matter system) induces selection rules as specified in the tables I and II in the text – there is no need to re-derive any theories for crystal structures - we build on top of the already known theory in condensed matter.
 - The main aim of the ~20 numerical examples in the SI (some of which were added following specific comments of the reviewers in previous rounds) is to demonstrate the validity of the analytical results. Indeed, beyond the numerical validation, each demonstration shows interesting features, etc., but they are NOT the purpose of the paper. We can of course add more numerical examples... yet it starts to feel like a never-ending process.
 - The main results of the paper are: introduction and formulation of dynamical symmetry Floquet group theory and its application to selection rules, using harmonic generation as a specific example, i.e. derivation of Tables I and II in the main text that present selection rules for ALL possible combinations of target/crystal point group, and laser dynamical group. These tables include NEW dynamical symmetries and selection rules, including elliptical, time-reversal, reflectional, non-collinear DSs, etc., which are all applicable to bulk crystals.
2. With respect to: “Stronger connections between the proposed time-dependent Hamiltonian approach and conventional polarization-dependent nonlinear optics in cases in which they coincide would be helpful in identifying the reasoning behind the conditions leading to divergence.”

It is unclear what the referee refers to. As we have shown, our theory fully reconstructs the specific examples that have been known from perturbative nonlinear optics for decades, as well as extending the analysis to novel cases. It is especially confusing that the referee mentions (but does not specify) some “divergence”, since our results fully reproduce previous derivations and experimental results. Can the referee name one selection rule in harmonic generation from bulk crystals (that is not due to propagation effects) that “diverges” from table II?
3. With respect to: “The utility of the time-dependent approach taken in this work is difficult to assess for interpretation of high harmonic generation. While harmonic generation in isotropic assemblies is reasonably straightforward, the enabling capability of the proposed approach is arguably the extension to assemblies of lower symmetry. Is this easy to do for a high harmonic from an assembly of low symmetry? How are the results to be interpreted?”

As explained in our reply to points 1 and 2 as well as in page 1 of this file, the theory is applied straightforwardly to harmonic generation from crystals.

4. With respect to: “This work describes a fairly specialized topic of polarization-dependent high harmonic generation from a purely theoretical perspective. As such, the work is likely to appeal to a relatively small fraction of the Nature Communications readership. It may be more appropriate to consider an alternative topical journal for publication, in which the target audience is more likely to come across the work.”,

We wish to point out that our theoretical group theory framework is most general and can be applied to any Floquet system (none of our proofs rely specifically on the HHG Hamiltonian). Thus, polarization dependent high harmonic generation is not the main topic of the paper, but only an example of the general theory and its many applications. For instance, such a theory would be useful for light-matter interactions in systems with many-body wave functions that cannot be solved for, such as BEC, Floquet topological insulators and more. Moreover, it will become an elementary tool in ultrafast spectroscopy, as we derive in this follow up work on chirality spectroscopy (<https://arxiv.org/abs/1807.02630>), but it applies more generally to magnetic interactions, spin-orbit systems, and more. We are confident this general theory will interest the broad readership of Nature Communications (as also acknowledged by the comments of referees #1 and #2).